# Understanding the Neurotrophic Virus Mechanisms and Their Potential Effect on Systemic Lupus Erythematosus Development

**DOI:** 10.3390/brainsci14010059

**Published:** 2024-01-06

**Authors:** Felipe R. Uribe, Valentina P. I. González, Alexis M. Kalergis, Jorge A. Soto, Karen Bohmwald

**Affiliations:** 1Millennium Institute on Immunology and Immunotherapy, Laboratorio de Inmunología Traslacional, Departamento de Ciencias Biológicas, Facultad de Ciencias de la Vida, Universidad Andrés Bello, Santiago 8370146, Chile; f.uribeleiva@uandresbello.edu (F.R.U.); v.gonzalezhenriquez@uandresbello.edu (V.P.I.G.); 2Millennium Institute on Immunology and Immunotherapy, Facultad de Ciencias Biológicas, Pontificia Universidad Católica de Chile, Santiago 8330025, Chile; akalergis@bio.puc.cl; 3Departamento de Endocrinología, Facultad de Medicina, Pontificia Universidad Católica de Chile, Santiago 8331150, Chile; 4Instituto de Ciencias Biomédicas, Facultad de Ciencias de la Salud, Universidad Autónoma, Santiago 8910060, Chile

**Keywords:** neurotropic viruses, central nervous system, aseptic meningitis, encephalitis, immune system, systemic lupus erythematosus

## Abstract

Central nervous system (CNS) pathologies are a public health concern, with viral infections one of their principal causes. These viruses are known as neurotropic pathogens, characterized by their ability to infiltrate the CNS and thus interact with various cell populations, inducing several diseases. The immune response elicited by neurotropic viruses in the CNS is commanded mainly by microglia, which, together with other local cells, can secrete inflammatory cytokines to fight the infection. The most relevant neurotropic viruses are adenovirus (AdV), cytomegalovirus (CMV), enterovirus (EV), Epstein–Barr Virus (EBV), herpes simplex virus type 1 (HSV-1), and herpes simplex virus type 2 (HSV-2), lymphocytic choriomeningitis virus (LCMV), and the newly discovered SARS-CoV-2. Several studies have associated a viral infection with systemic lupus erythematosus (SLE) and neuropsychiatric lupus (NPSLE) manifestations. This article will review the knowledge about viral infections, CNS pathologies, and the immune response against them. Also, it allows us to understand the relevance of the different viral proteins in developing neuronal pathologies, SLE and NPSLE.

## 1. Introduction

Infections of the central nervous system (CNS) are highly relevant worldwide due to their high mortality rate [1]. These infections can be caused by bacteria, fungi, protozoa, and viruses [1,2,3,4]. Since the beginning of the 20th century, it has been reported that several viruses, which are named neurotropic viruses, can infect the CNS [5]. 

The CNS pathologies associated with infections such as viral meningitis or encephalitis are highly relevant worldwide. The incidence of viral meningitis is higher than its bacterial equivalent [6]. A study in the United Kingdom indicated that the annual incidence of this pathology oscillates between 2 and 73 persons per 100,000 inhabitants [6]. On the other hand, the incidence of encephalitis has been decreasing over the decades. In 2019, it was estimated that this pathology had an incidence of 1,444,720 worldwide [7]. Both pathologies are characterized by similar symptoms, including fever, headache, nausea or vomiting, photophobia, seizures, malaise, and neck stiffness [7,8]. Reports have associated a considerable range of viruses with viral meningitis and encephalitis, including measles, henipaviruses, mumps, and flaviviruses, such as Japanese encephalitis, adenovirus (AdV), cytomegalovirus (CMV), enterovirus (EV), Epstein–Barr Virus (EBV), herpes simplex virus type 1 (HSV-1) and herpes simplex virus type 2 (HSV-2), lymphocytic choriomeningitis virus (LCMV), and severe acute respiratory syndrome coronavirus 2 (SARS-CoV-2) [8,9,10,11]. Some of these viruses are involved in developing autoimmune diseases, such as rheumatoid arthritis, type 1 diabetes, psoriasis, and systemic lupus erythematosus (SLE) [12,13,14,15,16,17,18]. However, they can also trigger CNS-associated autoimmune pathologies like multiple sclerosis [19,20]. Likewise, viral infections can trigger clinical manifestations associated with neuronal disorders in patients with SLE, called neuropsychiatric lupus (NPSLE) [10,11,12,13,14,15,21,22].

The CNS has neuroanatomical barriers that allow its protection, including the meninges, blood–cerebrospinal fluid barrier (BCSFB), the blood–brain barrier (BBB), and the fibroblasts from the olfactory nerve [23,24,25]. Additionally, it has been observed that immune cells can enhance these barriers by activating T cells with protective functions in response to a potentially neuroinvasive infection [26]. All this results in a complex structure that denies microorganisms access to the CNS from the nasal epithelium, cerebrospinal fluid (CSF), or via hematogenous spread [27,28,29]. Neurotropic viruses have mechanisms that allow them to invade the CNS, infect local cells, and induce a pro-inflammatory immune response that alters normal function [30,31,32].

This article will review the current knowledge on the main neurotropic viruses that have been clinically reported to cause aseptic meningitis due to the significant impact of this pathology [6]. Within this group, we find viruses such as AdV, CMV, EBV, HSV, and LCMV [10,11]. Additionally, we include SARS-CoV-2 due to its tremendous global relevance in recent years. We will also determine how some of them, especially CMV, EBV, and LCMV, can trigger SLE development and even some neuropsychiatric manifestations [12,14,33].

## 2. Neurotrophic Viruses

Neurotrophic viruses are pathogens that can cause alterations in the function of the CNS. These viruses can enter the CNS through the previously mentioned pathways. Additionally, it has been observed that viruses whose genome corresponds to RNA can be introduced into the CNS through nerves, such as the sciatic nerve [34]. Some can even transport immune cells, allowing them to arrive at the CNS [35]. On the other hand, dsDNA viruses can enter the CNS by penetrating the blood–brain barrier, utilizing brain microvascular endothelial cells, and even using sensory nerve endings and olfactory receptor neurons to enter the CNS [36]. When the infection occurs, the main mechanism that the CNS possesses to defend itself is the activation of microglia, which induce the activation of the innate immune response and a subsequent adaptive immune response [37]. Even the microglia can act with astrocytes to induce the chemotaxis of immune cells, like neutrophils, to combat infection generated in the CNS [38]. In this section, we will discuss the most relevant neurotropic viruses and the local immune response against them. 

### 2.1. Adenovirus

Adenovirus (AdV) is a viral agent capable of reaching the CNS. AdV is a member of the *Adenoviridae* family. This virus has a linear double-stranded DNA (dsDNA) genome (Table 1), varying from 25 to 45 Kb [39]. The viral particle has two main structural elements: the outer capsid and the core [40]. Within the core, the viral genome is associated with two primary proteins, polypeptide V and polypeptide VII, and a protein called mu (μ), which is rich in arginine [41]. These proteins function as DNA capacitors within the viral particle [41]. In addition, the outer capsid presents an icosahedral form, which is constituted by seven proteins (II, III, IIIa, IV, VI, VIII, IX), where II, III, and IV compose the hexon and penton of the viral structure [42]. On the other hand, researchers have identified other capsid proteins, such as IIIa, VI, VIII, which, together with μ protein, help stabilize the virion capsid [42]. Finally, also have been reported that VI and μ protein generate infectious viral particles, virus assembly, and viral transcription [43,44,45].

AdV can be found worldwide, and its associated infections can happen at any time of the year. These infections are usually unplanned and are more frequent in children and immunocompromised individuals [131]. When AdV infects the CNS, it results in neurological disorders such as febrile convulsions, encephalitis, acute disseminated encephalomyelitis, and meningitis (Table 1) [48]. AdV serotypes 1 to 7 have been reported in more than 80% of infections in children and adults in countries such as the United States, Canada, the United Kingdom, Taiwan, and South Korea [131]. In South America, on the other hand, serotype 7 is the most predominant [131]. Neurological manifestations caused by AdV are rare. It has been observed in pediatric patients that only 1.5% of AdV-infected children have neurological symptoms [132]. In addition, one study analyzed 48 cases of neurological manifestations caused by AdV in children, where 38% of them presented permanent neurological sequelae [133]. This virus can infect the CNS, and it can enter through the olfactory nerves, where the virus is transported through the axons of olfactory receptor neurons (ORNs) to the olfactory bulb (Table 1) [46]. Virus transport occurs retrogradely, where the virus is internalized at the synapse and then transported by the dynein motor complex into the cell body [134]. Additionally, AdV can cross the BBB by binding to the coxsackievirus receptor and the adenovirus receptor (CAR) located at the BBB tight junctions [47]. Thus, when the virus matures, it is released to the basolateral surface and can interact again with these receptors, disrupting the tight junction structures [47].. In pediatric patients diagnosed with AdV, 3.3% of cases have been found to cause febrile seizures, and there is a 0.4% incidence rate for encephalitis associated with AdV infection. Despite this, AdV is a rare cause of CNS disease, presenting with variable symptoms ranging from mild aseptic meningitis and reversible encephalopathy to severe acute necrotizing encephalopathy (Figure 1A) [133].

Infection with AdV activates innate immunity, producing pro-inflammatory and antiviral cytokines such as IFNs [135,136]. When the virus infects airway tracheal epithelial cells, it enters through fibers that bind to membrane receptors and adhesion factors [137]. This effect is enhanced by tumor necrosis factor (TNF)-α and interleukin (IL)-8; this last one also acts as a chemokine [137]. Both cytokines enhance the expression of the coxsackievirus receptor, CAR, and integrins, facilitating the entry of AdV into the host cells [137]. At the CNS level, *in vitro* experiments have shown that the AdV infection of microglia induces an increase in the expression of inflammatory mediators, especially inducible nitric oxide synthase (iNOS) and TNF-α [30]. This response is caused by the mitogen-activated protein kinase (MAPK) pathway, where the extracellular signal-regulated kinase (ERK) protein plays a relevant role [30]. The E1A protein plays a key role in viral infection, modulating transcription and suppressing the host’s innate antiviral responses [49]. This protein can alter the histone post-translational modifications associated with the latter function, specifically histone H2B [49]. E1A interacts with the hBre1 complex, preventing the occurrence of H2B monoubiquitination [49]. This modification is crucial in activating interferon-stimulated genes (ISGs) (Table 1) [49]. In addition, it has been noted that the binding of an E1A molecule to two proteins, namely forkhead box family transcription factor (FOXK) and C-terminal binding protein (CtBP), and gene *DCAF7* can effectively inhibit the activation of ISGs [138]. The virus can also modulate the immune response by changing the active state of the signal transducer and activator of transcription 1 (STAT1). It does this by blocking the dephosphorylation of STAT1, which occurs when the virus cannot interact with TC45 phosphatase. As a result, the virus can control the body’s response to IFNs (Table 1) [50]. In addition, AdV possesses proteins encoded in the early transcription region 3 (E3), whose functions can modulate the innate and adaptive immune response (Table 1) [51]. One identified protein is E3-gp19K, which can keep major histocompatibility complex (MHC)-I molecules within the endoplasmic reticulum. Furthermore, it hinders the function of the tapsin transporter, which is responsible for processing the peptides presented by MHC-I (Table 1) [52]. This mechanism suppresses signals over T cells, destroying cytotoxic T cells [52]. 

Clinical studies have extensively highlighted the neurological manifestations of Adenovirus (AdV) infections. This virus can breach the BBB, exploit the olfactory nerve, and infiltrate the CNS. Once inside, AdV utilizes proteins to effectively elude the antiviral immune response, triggering a cascade of clinical manifestations observed in affected individuals.

### 2.2. Cytomegalovirus

Another important neurotrophic virus relevant to SLE development is CMV, a member of the *Herpesviridae* family, whose genome corresponds to double-stranded DNA (dsDNA) with a size corresponding to 235 Kb (Table 1) [53]. The CMV genome can be divided into two main sections: unique long regions and unique short regions [53]. The virus’s envelope contains glycoproteins that play important roles in virus attachment, entry, maturation, and immune response evasion [139,140,141]. These include glycoproteins such as gB, gH, and UL16 [54,140,142]. Additionally, CMV encodes more than 200 ORFs, encoding 178 proteins. In addition, 32 proteins form the tegument, which plays different roles, from host cell conditioning at the onset of infection to the final stages of virion assembly [142,143].

CMV infections are widespread across the globe, particularly in developing nations, where approximately 90% of the population is affected [144]. CMV infections have a seroprevalence of 90% in the Eastern Mediterranean region and 88% in the Western Pacific and the African region [144]. Despite this, CMV is a virus that is generally asymptomatic once infection occurs, but in a smaller percentage, immunocompromised people can develop symptoms following infection [143]. However, CMV infection can trigger alterations at the CNS level, which are infrequent. However, in autopsy cases from immunocompromised patients, 18–28% of them have been affected with CMV-associated neurological manifestations [145]. CMV can over-regulate the activity of matrix metalloproteases (MMPs) throughout the brain, which can lead to BBB rupture via the degradation of the basement membrane or via the cleavage of the proteins occludin and claudin-5 from tight junctions (Table 1) [55]. Also, it is speculated that monocytes may be involved in BBB-associated passage. Once CMV crosses the BBB, it can infect resident cells [146]. The most prevalent clinical manifestations are polyradiculopathy (inflammation of the spinal nerve roots), myelitis, ventriculitis, dementia, and cranial nerve involvement (Table 1) [56,57]. Additionally, this virus can trigger congenital infection during gestation, directly affecting the CNS or late neurological sequelae such as sensorineural deafness (Figure 1B) [147]. The most severe manifestations associated with this phenomenon include microcephaly, periventricular calcifications, convulsions, spastic tetraplegia, and hydrocephalus [146]. On the other hand, it has been observed that 0.2–2% of newborns develop congenital CMV infection in utero, of which 10–15% have visceral organomegaly, microcephaly with intracranial calcifications, chorioretinitis, jaundice, mental retardation, sensorineural hearing loss (SNHL) and skin lesions [146]. All these conditions are because CMV can affect various cell populations of the CNS, such as astrocytes, neuronal cells, oligodendrocytes, and microglia [146]. Additionally, CMV infection can suppress the development of neural stem cells and negatively influence their differentiation (Figure 1B) (Table 1) [58].

During CMV infection, the innate immune response is facilitated by toll-like receptor 2 (TLR2), which identifies the surface glycoproteins gB and gH. This recognition triggers the activation of the nuclear factor kappa B (NF-κB)-dependent signal transduction pathway and the involvement of specificity protein 1 (Sp1) and interferon regulatory factor type 3 (IRF3) [54,148,149,150]. On the other hand, upon recognizing the presence of the CMV virion, the host’s immune system initiates an innate response by activating IRF7 and interferon-stimulated genes (ISG). This activation leads to the transcription of the ISG54 protein. Additionally, the host releases inflammatory molecules like chemokine ligand (CCL)5, IL-6, and IL-8 [151,152,153]. *In vitro* analyses have shown that CMV infection in microglia induces the increased secretion of inflammatory cytokines and chemokines, such as TNF-α, IL-6, CCL3, and CCL5 (Figure 1A) [154]. Even astrocytes secrete inflammatory mediators in response to CMV infection, including CCL2, IL-8, and CCL3 [154]. However, CMV can regulate the immune response through the infection of dendritic cells (DCs), which explicitly affects MHC. This event suppresses MHC-I, MHC-II, and co-stimulatory molecules such as CD40 and CD80 (Table 1) [59]. In addition, CMV can exert a masking effect through the UL16 glycoprotein to evade the response of natural killer (NK) cells [141]. Additionally, CMV has been observed to induce phenotypic and functional changes in macrophages, thereby increasing susceptibility to infection (Table 1) [60] and causing infected macrophages to lose the ability to present antigens because they lose transcripts and proteins associated with this function, such as the immune evasion genes m152, m4, and m6 (Table 1) [61].

While less prone to directly inducing CNS-related pathologies, CMV can rupture the BBB by triggering an overregulation of MMP activity. Nevertheless, its impact extends further as it can significantly affect fetal development, leading to congenital infections that hinder proper neuronal growth. This virus displays a versatile ability to target various cell types, including astrocytes, neuronal cells, oligodendrocytes, and microglia, thus ultimately contributing to the emergence of neurological manifestations. Notably, this virus has been related to the development of autoimmune diseases that affect the CNS, like SLE.

### 2.3. Enteroviruses

Other neurotrophic virus that have begun to have a growing incidence and relevance over time are enteroviruses (EVs), members of the *Picornaviridae* family. Their genome corresponds to monocistronic single-stranded positive-sense RNA (ssRNA^+^), with a length of 7.5 Kb (Table 1) [62]. EVs contain four structural proteins called VPs, which are numbered 1 through 4 and are part of the viral capsid [155]. Additionally, they have non-structural proteins, which promote viral infection and alter the development of the host antiviral response [63,156,157,158].

EVs can be found on every continent and infect people of all ages [159]. Geographically, Asia and Europe are the continents most affected by EVs. Worldwide, neurological manifestations associated with these viruses represent 29.4% [159]. EV infections are usually asymptomatic. However, it has been observed that four human EV species (EV-A to EV-D) are viruses that can trigger CNS-associated clinical conditions [160]. The EV-A and EV-B enteroviruses are the leading causes of CNS-associated manifestations, among which we find EV-A71, CVB5, and E30, among other species [159]. EV can enter the CNS through previously infected myeloid cells, which pass through the choroid plexus and penetrate the brain parenchyma (Table 1) [35]. These myeloid cells can be infected during extravasation into the choroid plexus and/or passage through the choroid plexus epithelium [35]. Similarly, EVs can enter the CNS through the BBB, using the endothelial cells that compose this barrier as an entry point and source of viral replication (Table 1) [64]. The EV-71 VP1 protein can increase BBB permeability by decreasing the expression of the tight junction protein claudin-5, leading to the induction of BBB leakage [65]. Furthermore, the virion can invade the brain parenchyma cellularly with elevated BBB leakage through the loosened tight junctions [65]. Additionally, it has been observed that EV-A71 infects motoneurons, especially at neuromuscular junctions, and thus can invade the CNS (Table 1) [66]. Once in the CNS, EVs can affect different cells, such as neural progenitor cells and astrocytes, and neurons have also been shown to be susceptible to EV infection [161]. This induces aseptic meningitis, encephalitis, and acute flaccid myelitis/paralysis (AFM) development (Table 1) [67]. EV-71, which is associated with the CNS, triggers a signaling cascade via the TLR9 receptor that induces the formation of IL-12p40, thus generating a negative effect that induces encephalitis (Table 1) [68]. IL-12p40 can damage neurons because it causes excessive neurotoxic nitric oxide (NO) production via iNOS [68]. The invasion of EV into the CNS can potentially cause cognitive impairments [69,70]. In infants as young as three months old, a decline in gross and fine motor skills, problem-solving abilities, socialization, and communication has been observed (Table 1) [69]. Similarly, it has been observed that patients who have suffered an EV-71 infection present a delay in neuronal development; in pediatric patients, a reduction in cognitive functioning was observed (Figure 1A) (Table 1) [70]. Notably, EV infection during childhood has been related to attention deficit hyperactivity disorder (ADHD) and epilepsy (Table 1) [71]. Furthermore, EV-D68 infection affects both inhibitory (GABAergic) and excitatory (glutaminergic) neurons, leading to a restructuring of the Golgi apparatus at the cellular level. This restructuring results in the formation of a replication organelle [162]. Additionally, the activity of neurons is reduced upon infection with EV-D68 [162].

In CNS EV-71 infection, cytokines such as IL-6, IL-8, and CXCL10/IP-10 have been observed in the CSF, which correlates with an increase in the recruitment of monocytes and neutrophils in this fluid [163]. Additionally, EV-71 infection in primary monocytes generates an immune response mediated by the secretion of IL-1, IL-6, and TGF-α [164]. In rat neuronal cells cultured *in vitro*, it has been observed that, in response to EV infection, both microglia and astrocytes can induce an inflammatory response through the secretion of NO, TNF-α, and IL-1β (Figure 1A) [165]. The absence of antiviral responses associated with type I interferon (IFN-I) may be because of the presence of the virus’s non-structural proteins in processes that enable the secretion of these cytokines (Table 1). The non-structural protein A2 of EV can perform this effect by suppressing the IFN-β gene [158]. Another mechanism described for strain EV-71 involves the suppression of the antiviral response, mediated by the non-structural protein 3C, which negatively regulates the cytosolic receptor retinoid acid-inducible gene I (RIG-I) pathway (Table 1) [63]. This effect demonstrates that IRF-3 cannot translocate to the nucleus and does not activate genes associated with the IFN-I response [63]. Additionally, EV-71 generates a negative regulation on RIG-I ubiquitination, inhibiting the pathway that allows IFN-I secretion; this causes the non-activation of ISG, among which we find ISG54 and ISG56 [63]. On the other hand, NLRP3 inflammasome activation also exerts a critical antiviral function against EV-71 because it allows the secretion of IL-1β [72]. In this line, when EV-71 replicates in myeloid cells, it can inhibit the inflammasome thanks to two non-structural proteins (2A and 3C), which can cleave the NLRP3 protein and inhibit IL-1β secretion (Table 1) [72]. Interestingly, the antiviral effects of the non-structural proteins described in EV-71 are observed to have similar functions in other virus strains, such as EV-68. In EV-D68, protein 3C excises IRF7, which induces a negative antiviral response mediated by IFN-I [73]. It has also been observed that the 2Apro protein from EV-D68, a protease, can act on tumor necrosis factor receptor-associated factor 3 (TRAF3), thereby inhibiting the production of IFN-I by the host (Table 1) [74]. 

Over time, enteroviruses have gained increasing prominence worldwide due to their significant role in the development of neurological manifestations. These viruses can breach the BBB by utilizing infected immune cells or motor neurons as a gateway into the CNS. Upon entry, they provoke an inflammatory response and cause cellular damage, resulting in distinct clinical manifestations associated with these infections. The development of autoimmune diseases like SLE and the possible link with this viral infection are poorly described.

### 2.4. Epstein-Barr Virus

Like CMV, Epstein–Barr Virus (EBV) is another *Herpesviridae* family member with great neurotrophic potential and is relevant in developing autoimmune diseases, especially SLE. The EBV genome corresponds to double-stranded DNA (dsDNA) with a length of around 172 Kb (Table 1) [75]. This virus contains seven structural proteins, which have fundamental roles for the virus; for example, BFRF3 is associated with the assembly of the viral capsid [166,167]. Several glycoproteins have diverse functions, such as the fusion of the viral envelope on the host cell, viral propagation, and the disruption of the antiviral response [168,169,170,171]. Additionally, this virus presents tegument proteins, whose functions are varied. They are associated with the release of mature virions, infectivity, viral production, and the inhibition of the host’s antiviral response [172,173,174,175,176].

It has been observed that EBV infections are not selective based on age, as they have the potential to impact individuals of all age groups, including both children and adults [177,178]. In China, it has been observed in recent years that in children who have suffered EBV infections, approximately 0.6% of them present neurological manifestations [179]. EBV infection can modulate the BBB because this virus infects microvascular endothelial cells. This induces inflammatory and endothelial markers that can trigger potential neurological manifestations (Table 1) [76]. Upon infection, the EBV dUTPase protein induces the expression of pro-inflammatory cytokines such as IL-6 and IL-1β in microvascular endothelial cells and microglia, as well as TNF-a in astrocytes [77]. These cytokines plus IFN-γ alter the integrity of the BBB by regulating the expression of genes encoding for proteins that are involved in forming and maintaining the tight junctions between endothelial cells in the capillaries of the BBB, as well as modulating cell adhesion and the extracellular matrix [77]. EBV can affect critical CNS cells, such as neurons and glial cells [180,181]. EBV can trigger encephalitis, in whose pathology CD4^+^ T and CD8^+^ T cells play an essential role by generating immune checkpoint inhibitors [9,182]. Other pathologies associated with the CNS induced by EBV are acute transverse myelitis, cerebellitis, ataxia, and Guillian–Barré syndrome (Figure 1A) (Table 1) [78,79]. When caused by EBV infection, cerebellar ataxia involves the presence of self-reactive antibodies (referred to as anti-neuronal antibodies) [183]. These antibodies are produced due to a phenomenon known as mimetism, where similarities between EBV proteins and neuronal antigens trigger their production [183]. On the other hand, EBV can trigger neurodegenerative diseases, including Alzheimer’s Disease (AD) and Parkinson’s disease (Table 1) [80,81]. An *in silico* analysis determined that EBV infection is associated with AD development due to the formation of peptide aggregates, specifically a 12-amino acid peptide derived from the glycoprotein gM (Figure 1A) [80]. Additionally, it has been observed that neuronal cells exposed to EBV and gM induce an environment that promotes neuroinflammation due to the positive upregulation of specific inflammatory cytokines such as IL-1β, IL-6, and TNF-α [31]. On the other hand, Parkinson’s disease has also been associated with EBV infection because it is caused by a molecular mimetism between the C-terminal region of α-synuclein and a repeat region in the latent membrane protein 1 (LMP1) protein from EBV [81]. 

It has been reported that EBV-infected patients show elevated levels of IL-6, IL-18, IFN-γ, TNF-α, and IL-10 in their serum [184]. At the CNS level, it has been observed in glial cell lines that, upon stimulation with EBV, these induce an increase in the expression of inflammatory cytokines such as IL-6 and IL-1β (Figure 1A) [180]. EBV possesses some mechanisms to evade the host antiviral response through different proteins. BILF1 inhibits NLRP3 inflammasome activation and subsequent pyroptosis, which inhibits viral replication (Table 1) [82]. This is because an interaction occurs between the BILF1 protein and the host UFL1 ligase, which generates UFMylations on mitochondrial antiviral signaling (MAVS), causing it to pack into vesicles (Table 1) [82]. On the other hand, EBV also induces an evasion pathway for IFN-I secretion (Table 1) [83]. Tyk2 phosphorylation is inhibited by cytoplasmic envelopment protein 2 (BGLF2), causing STAT2 and STAT3 not to be activated, resulting in the non-expression of IFN-I-associated genes [83]. The tegument protein, BFRF1, acts on IKKi, inhibiting its kinase activity; this blocks IRF3 translocation to the nucleus and inhibits the IFN-I pathway, specifically IFN-β (Table 1) [84]. Another tegument protein, BPLF1, has a deubiquitinase activity, which removes TBK1 ubiquitins while suppressing IRF3 dimerization [185]. Additionally, BKRF4 has immune response inhibitory activity since this protein interacts with histones H2A-H2B, causing EBV to have a mimetic effect since the host cannot detect the signals in response to DNA damage [173]. Also, the EBV-encoded miR-BART16 can suppress the production of IFN-stimulated genes and hinder the anti-proliferative impact of IFN-α on B cells during latent infection (Table 1) [186]. On the other hand, it is known that EBV can infect monocytes through a gp25–gp42–gp85 complex, which interacts with the membrane of this immune cell, triggering apoptosis and inhibiting the development of DCs (Table 1) [85]. Likewise, it has been observed that the glycoprotein gp42 can interact with human leukocyte antigen (HLA)-DR1 of MHC-II, promoting B cell infection [187]. EBV infection over B cells can generate reprogramming on them, decreasing CXCR4 expression, modulating CD23 expression, and even regulating more than 11,000 genes associated either metabolically or phenotypically (Table 1) [86,87,88].

As we described earlier, the relevance of EBV is due to its modulatory effect on the cells of the immune system. This virus can enter the CNS through the BBB and then interact with neurons and glial cells, triggering an inflammatory and tissue-damaging process that leads to CNS pathologies. Notably, the role of EBV in autoimmune diseases such as MS and SLE makes the timely diagnosis and development of new treatments to prevent these diseases necessary.

### 2.5. Herpes Simplex Virus

*Herpesviridae* family members possess several viruses capable of developing neurotrophic conditions, such as CMV and EBV. In addition, we can find the herpes simplex virus (HSV), whose genome is a double-stranded DNA (dsDNA) of approximately 150 Kb (Table 1) [89]. This virus has many proteins, among which we find tegument proteins and glycoproteins associated with virion assembly, DNA encapsidation, and egress [188]. In addition, some glycoproteins, such as gC and gE, are related to the modulation of the host’s immune response [90,91].

HSV infections are responsible for most genital and neonatal infections globally, accounting for approximately 85% of cases [189]. Additionally, there has been a rise in the age at which individuals first engage in sexual activity [189]. HSV infections are most prevalent in Africa and America [190]. HSV can affect the CNS via local spread from the periphery or viremia. In the case of HSV-1 infections, there are believed to be three routes by which it can infect the CNS [92]. The first is from the primary oropharyngeal infection site, which can reach the brain via the trigeminal or olfactory nerves. The second involves the same neural pathways but is caused by a reactivation of initial infection in the periphery and, finally, the reactivation of latent HSV-1 in situ in the brain (Table 1) [92]. The trigeminal and olfactory nerve pathways allow HSV to infect the CNS by circumventing cellular barriers, such as the blood–brain and cerebrospinal fluid barriers, leading to infection (Table 1) [93].

On the other hand, HSV-1 infection triggers Golgi stress through changes in morphology and function. The GM130 protein is downregulated, and there is an increase in apoptotic cells, regulating the stress response of the Golgi apparatus. This contributes to alterations in the structure and function of the BBB by downregulating occludin and claudin-5 [94]. In addition, Golgi apparatus fragmentation after infection leads to increased Ca^2+^ release, stimulating endothelial NOS (eNOS) activation and modulating the NO and vascular endothelial growth factor (VEGF) effectors that lead to BBB rupture [95]. When HSV infects the CNS, several cells are affected, mainly neurons, but astrocytes and oligodendrocytes have also been susceptible to HSV infection [191]. Despite this, the exact mechanism by which HSV-1 can destroy neuronal cells is unknown. However, this could be due to direct injury caused by the virus or immune-mediated cell injury [93]. HSV can induce meningitis and acute encephalitis, affecting individuals of all age ranges (Figure 1A) (Table 1) [10,11,96]. Encephalitis caused by HSV is uncommon since it has been observed to have an annual incidence rate estimated at 1 per 250,000–500,000 inhabitants [192]. The principal causative agent is HSV-1 [192]. Additionally, it has been observed that HSV-1 is present in 2.8% of Korean patients with encephalitis [193]. On the other hand, HSV-2 is found in 5.7% of patients with aseptic meningitis [193]. Encephalitis caused by HSV-2 affects glial cells, causing hemorrhagic necrosis in the temporal lobe [96]. Studies suggest that the epinephrine and corticosterone receptors expressed on sensory and sympathetic neurons modulate the potential for HSV infection [96]. Furthermore, the infection of the CNS with HSV-1 can result in cognitive impairment, although it does not reach the severity of dementia or other neurodegenerative conditions such as AD (Table 1) [97]. 

HSV infection triggers an immune response associated with pathogen-associated molecular patterns (PAMPs) through the TLR pathway, activating different proteins; among these are RIG-1, melanoma differentiation-associated protein 5 (MDA5), cyclic GMP-AMP synthase (cGAS), interferon-inducible protein AIM2 (AIM2), interferon-gamma-inducible protein 16 (IFI16), and DEAH-box proteins (DHX) [194,195,196]. This pathway initiates intracellular signaling pathways that trigger the production and release of molecules that hinder viral replication [194,195,196]. Additionally, the presence of TLR2 in antigen-presenting cells (APCs) activates NF-κB, which induce inflammatory cytokines production such as TNF, IL-1, IL-6, IL-8, IFN-γ, CXCL5, CXCL9, CXCL10, and CCL3 [197,198,199,200]. Microglia play a fundamental role in combating HSV infection in the CNS because they induce antiviral responses mediated by cGAS-STING, triggering IFN-I expression [201]. In astrocytes, infection with HSV-1 induces this cell population to secrete CXCL1, which causes the increased migration of neutrophils to the site of infection [202].

Like other viruses, HSV can modulate the host’s antiviral response. HSV-2 inhibits the ability of the cells it infects to recognize viral dsRNA, causing changes in the IFN-I pathways (Table 1) [98]. The early viral protein ICP0 can also inhibit IRF3, blocking the transcription of target genes associated with this factor (Table 1) [99]. On the other hand, to prevent the activation of protein kinase R (PKR), HSV-1 utilizes the late genes γ34.5 and Us11 to hinder the activation of this protein. As a result, the translation of eIF2A remains inactive, leading to the inability to suppress the translation of viral mRNAs [203]. The glycoproteins from EBV can also modulate the immune response, where gC binds to the C3b component of the complement and prevents the interaction of C5 and properdin with C3b. As a result, it hinders the activation of both the classical and alternative pathways of complement activation (Table 1) [91]. Moreover, HSV can regulate the adaptative immune responses. The glycoprotein gE hinders the ability of IgG antibodies to carry out their intended function by attaching to the Fc section (Table 1) [90]. HSV-2 can impact DCs, leading to their apoptosis and hindering the activation of T cells (Table 1) [100]. Furthermore, it has been observed that HSV-1 utilizes the γ34.5 protein to hinder autophagosome development within infected cells. As a result, the ability to present antigens to T cells is impeded, disrupting their regulation [204]. The effect associated with the antigens present is HSV is accomplished using a VHS-RNase. This enzyme facilitates the breakdown of host mRNA, decreasing the production of antiviral components [101]. The combination of these factors, along with the presence of the ICP47 protein, leads to a reduction in the expression of MHC-I on the surface of cells and a decrease in MHC-II levels. This impairment affects the ability to present antigens and ultimately hinders cellular and humoral immune responses (Table 1) [101,102,103,104,205].

Since HSV infection is one of the most common worldwide, the ability of this virus to trigger CNS-associated manifestations is concerning. HSV can use the trigeminal and olfactory nerves to invade the CNS and, by stressing the Golgi apparatus, can modulate the BBB’s structure. By targeting cells such as neurons, astrocytes, and oligodendrocytes, this virus provokes neurological pathologies. Recent findings associate HSV viral infection with autoimmune disease, but little is known in this field, and more research is needed.

### 2.6. Lymphocytic Choriomeningitis Virus

In contrast to the mentioned viruses belonging to the *Herpesviridae* family, the LCMV virus belongs to the *Arenaviridae* family, which is characterized by being an enveloped virus with a bi-segmented negative-sense single-stranded RNA (ssRNA^−^) genome (Table 1) [105], which corresponds to a short segment (S) of 3.5 Kb and a long segment (L) of 7.2 Kb [105]. The RNA S section encodes for highly relevant structural proteins, specifically nucleoproteins and glycoproteins [206]. On the other hand, the RNA L section codes for the L protein corresponding to the RNA polymerase of the virus. The N-terminal region acts as an RNA endonuclease [207,208,209]. Additionally, this genomic segment codes for a protein containing a zinc fingers sequence, which may be associated with RNA binding with a regulatory function [210]. The LCMV nucleoprotein (NP) is involved in processes that facilitate the intracellular spread of the virus, transcription, translation, and genome replication. Furthermore, it acts on host immune responses [211,212,213,214,215]. LCMV glycoproteins are associated with receptor-binding infection persistence and are related to virus fusion [216,217].

The primary reservoir for LCMV is rodents, particularly mice [218]. Viral human infection is due to direct contact with infected animals [218]. Over the years, this virus has been found in several countries worldwide because LCMV is prevalent in American, African, Asian, and European mice [219]. Studies performed in the last decade in Iraq and Finland have indicated that 5.1% and 5%, respectively, of people who have been affected by a neuroinvasive disease are positive for LCMV [219]. LCMV can infect the ependyma, choroid plexus, and brain meningeal regions. All this leads to LCMV infection generating BBB alteration (Table 1) [106]. CD8^+^ T cells are relevant to combat LCMV infection. However, this lymphocyte population can modulate the BBB through perforin secretion or by secreting chemokines that recruit monocytes and neutrophils, inducing tissue damage [107,108,109]. In a murine model, especially in rats, it has been observed that astrocytes and Bergmann glial cells are the main brain parenchymal cells infected by LCMV. The virus can spread through these cells, specifically neurons in the cerebellum, olfactory bulb, dentate gyrus, and periventricular region [220]. Individuals infected with LCMV often have CNS-associated problems, such as aseptic meningitis (Table 1) [10]. The symptoms associated with developing LCMV infection are varied and may include fever, myalgia, headache, and malaise [221]. However, some patients develop photophobia, vomiting, and nuchal rigidity [221]. Once LCMV infection of the CNS occurs, cells such as neurons and astrocytes are affected [222]. On the other hand, when a woman acquires LCMV infection during the second trimester of pregnancy, it has been observed that this infection can lead to the development of congenital disabilities in the fetus [110,111]. These defects may include hydrocephalus and chorioretinitis (Figure 1B) (Table 1) [110]. Also, when pregnant women contract LCMV, their babies can develop conditions such as macrocephaly, microcephaly, chorioretinopathy, and in some cases, experience severe neurological effects like spastic quadriparesis, seizures, visual loss, or mental retardation (Figure 1A) (Table 1) [112]. These congenital effects are due mainly to the fact that LCMV infection affects neuroblasts, which are calcified in the brain’s periventricular region by congenital infection [223]. In addition, LCMV alters neuron migration [223]. Also, murine models, particularly rats, have demonstrated the occurrence of neurodegenerative manifestations (Table 1) [224]. LCMV-infected rats have a movement of T cells toward the brain [224]. This migration serves to control glial infection while also contributing to brain degeneration. Notably, this degeneration does not affect the hippocampus, an area of the brain highly associated with severe neuropsychiatric diseases [202].

The activation of the innate immune response against LCMV infection is linked to the TLR2 and MyD88 pathways [225]. These pathways are responsible for triggering the IFN-I production, particularly IFN-α [225]. At the same time, MyD88 is associated with activating antiviral CD8^+^ T cells to take antiviral action against LCMV [225]. The secretion of IL-27 by B cells is regarded as having antiviral properties, as it facilitates the survival of the CD4^+^ T cells that target LCMV and encourages antibody class switching [226]. In a study conducted on mice, it was demonstrated that there is a distinct immune response to LCMV infection in males and females [227]. The CSF from males has lower levels of CCL5, IL-1α, and IL-1β compared to females and has a lower activation of APCs [227]. By the eighth day after infection, males exhibit decreased levels of IFN-γ compared to females. During this stage of viral infection, the production of this cytokine relies on T cells. Therefore, this observation suggests a potential disruption in the functioning of T cells in males [227]. The relevance of IFN-γ in the CD4^+^ T response against LCMV in the CNS is significant, as this cytokine plays a crucial role in inducing MHC-II expression [189]. Even DCs, in response to infection, can secrete inflammatory cytokines, IL-12 and IL-23, through TLR activation [228]. In the CNS, in response to LCMV infection, both microglia and astrocytes can produce inflammation-associated molecules such as TNF-α, CCL5, and CCL2 (Figure 1A) [229]. Additionally, glial cells recognize LCMV through TLR2, thus activating the MyD88 pathway [229]. However, LCMV has mechanisms to evade the antiviral response. The NP protein plays a crucial role in suppressing the production of IFN (Table 1). Its exoribonuclease activity removes viral RNA, prevents phosphorylation IRF3, and does not migrate to the nucleus, suppressing IFN-I production [113]. Additionally, the NP protein can interact with the kinase domain of the IKKε protein, preventing it from phosphorylating the IRF-3 protein. Consequently, this interaction hampers the production of IFN-I (Table 1) [114].

LCMV infection has shown its ability to trigger CNS-associated pathologies, where CD8^+^ T cells play a crucial role in modulating the BBB. However, this virus not only affects children and adults but can also affect fetuses through congenital infection, principally affecting neurons and interfering with the correct neuronal development, generating severe and irreversible pathologies. The relationship between the infection of LCMV and its role in autoimmune diseases like SLE is still understood.

### 2.7. Severe Acute Respiratory Syndrome Coronavirus-2

Unlike all the neurotrophic viruses mentioned in this section, SARS-CoV-2 has significantly impacted health worldwide in recent years and is mainly characterized by the respiratory disease it triggers. The recently described SARS-CoV-2 is a member of the *Coronaviridae* family, which is characterized by a single-stranded RNA genome that is positive-sense (ssRNA^+^) and non-segmented (Table 1). The size of this genome can fluctuate between 27 and 32 Kb [115,116]. In contrast, the length of SARS-CoV-2 specifically falls within the range of 29.8 to 29.9 Kb [230]. This virus encodes for four structural proteins: spike protein (S), envelop protein (E), membrane protein (M), and nucleocapsid protein (N) [231,232,233]. Additionally, the non-structural proteins (NSPs) that present these proteins play crucial roles in virus pathogenicity, genome replication, and the modulation of the immune response [117,234,235]. Finally, accessory proteins are relevant in evading the immune response [118,119,236,237,238,239].

The SARS-CoV-2 virus caused a global health crisis, where no country or continent was exempt from suffering its effects. It has been observed that socioeconomic status is a factor that increases the incidence of this virus; in sectors that have a lower socioeconomic status, there is a higher SARS-CoV-2 positivity rate compared to sectors that have a better status [240]. On the other hand, SARS-CoV-2-hospitalized patients who previously did not have a neurological disease frequently had CNS-associated manifestations (approximately 33%) [241]. This virus can enter the CNS because it can cross the BBB by altering the tight junctions (Table 1) [120]. This occurs because the virus can modulate the expression of MMP9, an endopeptidase necessary for remodeling the extracellular matrix, causing collagen IV degradation [120]. Also, it has been observed that components of the spike protein, such as S1 and S2, cause BBB leakage induction [121]. Additionally, SARS-CoV-2 can interact with cranial nerves IX (glossopharyngeal nerve) and X (vagal nerve) in such a way that it can potentially enter the CNS via this pathway; this is because the ACE-2, NRP1, and TMPRSS2 proteins are found in these cranial pairs (Table 1) [122].

Astrocytes are the primary target of SARS-CoV-2 infection in the CNS, as the virus utilizes the NRP1 receptor to enter these cells and establish itself as a site for viral replication [32]. Astrocytes infected by this virus have decreased glutamine metabolism intermediates (glutamate and GABA) [242]. In addition to NRP1, astrocytes possess other molecules that allow the SARS-CoV-2 virus to enter the target cells. These additional molecules include TMPRSS-2 and ACE-2 proteins [32]. The presence of the virus in the CNS causes neuronal symptoms by inducing neuron–neuron or neuron–glia fusion [243]. The process of neuronal fusion occurs sequentially and, after that, significantly impairs neuronal function [243]. Once SARS-CoV-2 interacts with the CNS, it can generate pathologies such as encephalitis and encephalopathy, seizure and epilepsy, and even neurodegenerative diseases, such as AD (Figure 1 A) (Table 1) [123,124,125,126]. It has been noted that SARS-CoV-2 has the potential to contribute to the development of AD. This observation is supported by the fact that viral infection causes modifications in gene expression, particularly in genes linked to the Wnt pathway, which is known to be disrupted in this neurodegenerative condition [123]. There are additional pathways that could contribute to the progression of this disease. These pathways include changes in inflammation, the processing of amyloid proteins, cellular trafficking linked to the transport of proteins in the endoplasmic reticulum, the breakdown of proteins involved in ubiquitin, and the regulation of calcium levels [123].

It is widely recognized that SARS-CoV-2 induces a cytokine storm, leading to an increase in inflammatory cytokines such as IL-8, IL-15, IFN-α2a, IFN-γ, CXCL10, CCL2, and TNF-α [244,245]. Regarding transcription, there is an increase in the activation of inflammatory pathways associated with NF-κB and IL-6-STAT3 [246]. Simultaneously, the E protein of SARS-CoV-2 interacts with TLR2, initiating a signaling pathway that involves NF-kB. This activation enables the transcription of inflammatory cytokines and chemokines, such as CXCL8, which share a similar context [247]. Likewise, the ORF7b protein promotes IFN-I signaling and the expression of inflammatory cytokines such as TNF-α and IL-6 [238]. In individuals with CNS infection caused by SARS-CoV-2, inflammatory cytokines like IL-4, IL-6, and IL-12 are released by microglia and astrocytes (Figure 1A) [125]. On the other hand, SARS-CoV-2 can modulate the host’s immune response and evade antiviral responses. The N protein affects RIG-I recognition on dsRNA (Table 1) [127]. Additionally, it has been observed that both the N protein and NSP5 protein can attenuate the formation of antiviral granules and suppress the IFN pathway (Table 1) [127,128]. The mechanism of inhibition in this pathway is related to the inhibition of TBK1 and IRF3 phosphorylation, resulting in the impossibility of IRF3 translocating to the nucleus [127]. The non-structural protein NSP5 can cleave the essential modulator NF-κB (NEMO), a relevant kinase for the RIG-I pathway (Table 1) [129]. Another non-structural protein, NSP12, affects the response to IFN by preventing the nuclear translocation of IRF3 (Table 1) [117]. The C-terminal segment of the ORF3b protein from the SARS-CoV-2 virus can reduce the activity of IFN-I by inhibiting the movement of IRF3 to the nucleus [236]. The binding of ORF6 to STAT1 prevents phosphorylated STAT1 from migrating to the nucleus, inhibiting antiviral responses and increasing SARS-CoV-2 viral replication [237]. On the other hand, ORF10 can act on the signaling pathway associated with MAVS because this accessory protein can induce MAVS autophagy, thus inhibiting the IFN-I response and promoting viral replication simultaneously (Table 1) [119]. This virus can also module the immune chemotaxis because the N protein can bind to these molecules, including CXCL12B, CCL8, CCL26, and CXCL10 (Table 1) [130]. Additionally, ORF7a generates a molecular mimetic of β2-microglobulin (β2m) that can interact with MHC-I, inducing the antigen presentation processes and, consequently, the antiviral response (Table 1) [118].

In contrast to the significant respiratory impact caused by SARS-CoV-2 globally, there has been a recent recognition of its relevance in developing neurological diseases. These conditions arise from the virus’s invasion of the CNS through the BBB and cranial nerves. Simultaneously, this invasion triggers an evasion of the antiviral immune response, leading to substantial damage to neuronal cells. The role of this virus in developing autoimmune diseases remain to be elucidated.

Each virus described is characterized and associated mainly with pathologies not directly linked to the CNS. However, as we have described, each of these viruses can invade the CNS in different ways, where two mechanisms are the most relevant for their infection; these include BBB modulation and entry through nerves [48,55,65,92,121,122]. These viruses, once inside this system, can interact and infect various cells, such as astrocytes, which are one of the most abundant cell types in the brain; they play a neuronal support role and can participate in synapses [161,180,181,220,242,243]. At the same time, this infection process leads to the development of an innate response by immune cells specific to this tissue, such as microglia, and a subsequent adaptive response [30,54,125,154,165,180,201,225]. All of this is to eliminate the virus. However, each virus can evade the host’s immune response through its different proteins. This leads principally to the virus not being eliminated and the generation of neurological clinical pictures and even neurodegenerative diseases [48,50,56,57,59,67,71,72,80,81,84,96,99,110,113,119,123].

## 3. Viral Infections Could Trigger SLE Development

Each virus mentioned in the previous section has a possible role in autoimmune disease development since these pathologies result from the intricate interplay between genetic factors and environmental influences [248]. Viruses are an environmental factor that is highly relevant as a possible trigger for the development of autoimmune diseases [249]. Some of the above viruses are involved in developing CNS-associated autoimmune diseases. Specifically, CMV and EBV are involved in the development of multiple sclerosis [19,20,250]. However, many of them are also involved in the onset and development of SLE and a neuronal condition in patients with SLE known as NPSLE [12,13,14,251,252].

SLE is characterized by the loss of immune tolerance, affecting T and B cells [253,254]. This phenomenon triggers the production of autoantibodies and pro-inflammatory cytokines, including IFN-γ, IL-6, IL-23, and IL-17, among other molecules [255,256,257,258]. This results in tissue damage, compromising the integrity of the different tissues of individuals affected by this pathology [259,260]. This condition possesses the capability to impact all the organs that are encompassed by the other autoimmune disorders previously mentioned. The global annual incidence of SLE varies between 1.5 and 11 individuals per 100,000 population [261]. Moreover, the prevalence of this pathology varies between 13 and 7713.5 persons per 100,000 inhabitants per year [261].

There are instances in which CMV infections have been reported to trigger SLE in patients without any prior history of the disease. These cases involved individuals who developed the clinical parameters associated with SLE after being infected with CMV [14]. Specifically, these are anti-nuclear antibodies, anti-dsDNA, anti-Sm/RNP, anti-cardiolipin, rheumatoid factor, decreased complement levels, and proteinuria (Figure 2A) [14]. Furthermore, researchers have noted the presence of 11 viral genes linked to CMV infection in the peripheral blood mononuclear cells (PBMC) of SLE patients. These genes include *UL32*, *UL36*, *UL44*, *UL50*, *UL56*, *UL82*, *UL84*, *UL95*, *UL105*, *UL117*, and *US31* (Figure 2A) [262]. This last one is highly expressed in PBMC and associated with inflammatory phenotypes, allowing M1 macrophages to be activated [262]. The involvement of the CMV UL44 protein in developing SLE is suggested due to its ability to co-precipitate with nuclear autoantibodies such as nucleolin, dsDNA, and ku70 [263]. On the other hand, in the NZB/W murine model, it has been observed that CMV phosphoprotein 65 (HCMVpp65) is associated with the formation of autoantibodies [264]. This murine model is considered a spontaneous SLE model characterized by the most characteristic clinical symptoms of SLE patients, including anti-nuclear and anti-dsDNA antibody formation, glomerulonephritis, and mild vasculitis. However, this model does not exhibit clinical manifestations such as skin rash or arthritis. It should be mentioned that, as in humans, this model is characterized by a higher prevalence in female mice than in male mice [265]. Previous studies have reported that BALB/c mice immunized either with the pp65 antigen or with a fragment in conjunction with Freund’s adjuvant-induced the production of antibodies against this antigen transiently [264]. However, when this antigen is administered together with the C3d adjuvant, which generates a stronger and more sustained immune response over time, non-autoimmune BALB/c mice produce autoantibodies such as anti-chromatin, anti-centriole, anti-mitotic spindle type I/II (MSA I/II), and anti-dsDNA. These findings suggest that this virus may play a role in initiating SLE (Figure 2A) [264]. SLE murine models are important because they allow us to understand what happens in humans during the pathology [265]. These models present the same clinical characteristics as a human. Additionally, they allow us to understand the triggered pathological mechanisms and even test new therapies [265].

In patients with SLE, the levels of IgM EBV antibodies are considerably elevated compared to those found in healthy individuals [13]. In the same manner, it has been noted that individuals with SLE have a significantly higher EBV viral load in their PBMCs compared to those who are in good health [266]. EBV can modify genes, and ten specific genes are closely linked to the progression of SLE. These genes include *CD40, LYST, JAZF1, IRF5, BLK, IKZF2, IL12RB2, FAM167A, PTPRC,* and *SLC15A* (Figure 2B) [33]. These genes are modulated through the EBNA2 protein belonging to EBV (Figure 2B) [33]. Moreover, the EBV LMP2A protein can modulate B cell differentiation and functionality [267]. This effect occurs because this protein induces an anti-Sm response by B cells and increases the TLR response shown by this lymphoid population, such that these effects can modulate the presence of autoreactive B cells, which has a direct implication in SLE (Figure 2B) [267]. It is important to mention that the EBNA1 EBV protein can stimulate the generation of anti-dsDNA antibodies (Figure 2B). These antibodies can accumulate in the kidneys and lead to renal damage. This renal damage may be connected to the kidney issues commonly observed in individuals with SLE [268].

LCMV can influence the development of SLE. Studies have shown that when C57BL/6 mice are infected with LCMV during the early stages of life and later exposed to crystalline silica, they produce autoantibodies associated with SLE (anti-chromatin and anti-Sm/RNP) (Figure 2C). Furthermore, these mice also exhibit renal damage [12]. Additionally, this virus can modulate the onset of SLE thanks to the activation of TLR and plasmacytoid dendritic cells (pDC) [269]. TLR-triggered signaling is crucial for developing LCMV-mediated SLE in the NZB lupus murine model [269].

The nervous system is one of the organs that can be affected in patients with SLE, known as NPSLE. This pathology has a 2% to 91% prevalence and is also an important source of morbidity in these patients, being the second most significant cause of mortality after lupus nephritis [270]. The clinical symptoms that occur during the development of NPSLE include headache, mood disorders, anxiety, mild cognitive dysfunction, cerebrovascular disease, seizure disorders, acute convulsive states, and neuropathies [271]. These symptoms develop due to the vascular, BBB, and brain parenchymal lesions. Studies have shown that autoantibodies can mediate damage, leading to focal or diffuse CNS effects [272,273].

The formation of the autoantibodies generated during viral infections triggers the development of some clinical pictures associated with NPSLE. This effect can be seen in CMV infection, which induces the formation of antibodies against cardiolipins (anti-phospholipid) (Figure 2D) [14,252]; these are associated with vascular endothelial cell injury, platelet activation, and thrombosis [252]. These effects may induce clinical symptoms of NPSLE, such as headache, stroke (cerebral ischemia and/or intracranial embolism), epilepsy, and cognitive dysfunction (Figure 2D) [252]. Also, the formation of anti-Sm antibodies induced by CMV, EBV, and LCMV infections (Figure 2D) [12,14,267] has been described. This may also be associated with the induction of NPSLE because this antibody has been demonstrated in a culture of neuroblastoma lines that can react against these cells, thus inducing neurotoxicity; this is associated with the development of an acute confusion state, which is one of the manifestations of NPSLE [251].

On the other hand, the viruses described in this article include AdV and HSV. They do not directly relate to SLE’s development or progression. However, people with this autoimmune pathology are more susceptible to infection by any of these viruses [274,275,276]. On the other hand, EVs similarly do not induce SLE. However, it has recently been observed that enterovirus coacksakie B4 can produce systemic mimetics like SLE. Because it can produce SLE-like skin rashes (CSLE), if proper clinical testing is not performed, it could lead to misdiagnosis in the future [277].

Since SLE has no defined etiology, viruses are highly relevant environmental factors in its development. Among these, we have observed that CMV, EBV, and LCMV can modulate genes that induce the development of this pathology through various molecular mechanisms. Additionally, they can induce SLE and CNS-associated SLE symptomatology, known as NPSLE. We also described that infection by some viruses can potentiate the production of autoantibodies, which ultimately trigger or potentiate all the clinical symptoms associated with NPSLE. Therefore, viruses are multifactorial elements capable of inducing both the base pathology and a subtype of it.

## 4. Conclusions

Neurotropic viruses are present in several CNS-associated pathologies, as mentioned throughout the article; this is reflected in Table 1. RNA and DNA neurotrophic viruses can infiltrate the CNS through various pathways, inducing an innate immune response triggered by CNS-resident cells, such as microglia and astrocytes; this induces the secretion of various inflammatory mediators [30,125,154,165,180,202,229]. It should be noted that some of the viruses mentioned, specifically CMV and LCMV, can cause congenital infections that will affect the correct neuronal fetal development [110,147]. On the other hand, these viruses use several evasion mechanisms to evade the antiviral immune response, including IFN response pathway modulation, antigen presentation process alteration, and lymphoid response modulation [52,59,63,84].

Since the triggering of CNS-associated pathologies does not primarily characterize each virus mentioned in the article, they have gained relevance over the years. It is therefore necessary to investigate each of these viruses in more detail to better understand how they can generate potential CNS infections and infections associated with other systems and/or tissues of our organism. It was considered that certain viruses, especially *Herpesviridae* family members, can modulate genes and even serve specific genes within the genome of a host cell, such as CMV or EBV [86,87,88,262]. Next-generation sequencing (NGS) techniques allow us to diagnose patients and know which virus affects them [278]. It also opens the possibility of better understanding what is modified at the transcriptional level during the pathology, allowing us to understand the diseases better. Even more advanced technologies, such as single cells, would give us a picture of what happens in a single cell at the transcriptional level [279] and potentially contribute to determining what occurs not only with neurons but also with epithelial tissue and even immune cells. This would give us an overall picture that can be connected to understanding the pathology in question better. All this will bring us closer to finding future treatments for these pathologies; for example, using nanoparticles to deal with neurotrophic virus infections has been observed [280,281]. Considering all this, understanding what is happening with each cell to elucidate both a general and specific overview is highly relevant for the future when aiming to obtain specific targets for the treatment of the various neurological pathologies discussed in this article, from meningitis to neurodegenerative diseases such as AD.

At the same time, as this information is being studied, the diagnostic techniques used for neurotrophic viruses are becoming more sophisticated; they are a tremendous clinical option for using NGS to provide more accurate treatments and even determine whether a patient is suffering from an infection caused by more than one pathogen. Therefore, it is relevant to study new therapies that are potentially highly effective against neurotrophic viruses, as has been observed with nanoparticles.

On the other hand, some of these viruses are also implicated in developing CNS-associated autoimmune diseases, namely SLE and NPSLE. Some viruses, such as CMV, EBV, and LCMV, can contribute to the progression of the pathology by modifying several genes associated with the development of SLE and increasing the production of autoantibodies, which may ultimately trigger the development of NPSLE [33,262,269]. This leads us to speculate that treatments against viral infections may be viable for patients with SLE who do not suffer from neuropsychiatric symptoms. However, it has been observed that drugs such as Acyclovir (ACV) and Valacyclovir (VACV), used in these instances, can trigger side effects that impact the CNS, in addition to nephrotoxicity, which is extremely dangerous considering that SLE patients usually suffer from glomerulonephritis [282]. Considering all this, it has been observed that immunotherapy using CD8^+^ T lymphocytes is regarded as a promising prophylactic method against these infections that, at the same time, does not generate secondary effects on the organism [282]. Thus, immunotherapy could be an excellent ally for these patients in preventing clinical manifestations of the CNS associated with viral infections. Considering all this, it is important to continue the research in this field to allow the development of new and effective strategies to control these pathologies.

All this gives us the understanding that viruses have great versatility in triggering acute and chronic pathologies. Due to their multiple components that can modulate the different cells of our organism, viruses can also modulate genes to trigger SLE. Similarly, other viruses can modulate genes to trigger SLE and its neuropsychiatric symptomatology.

## Figures and Tables

**Figure 1 brainsci-14-00059-f001:**
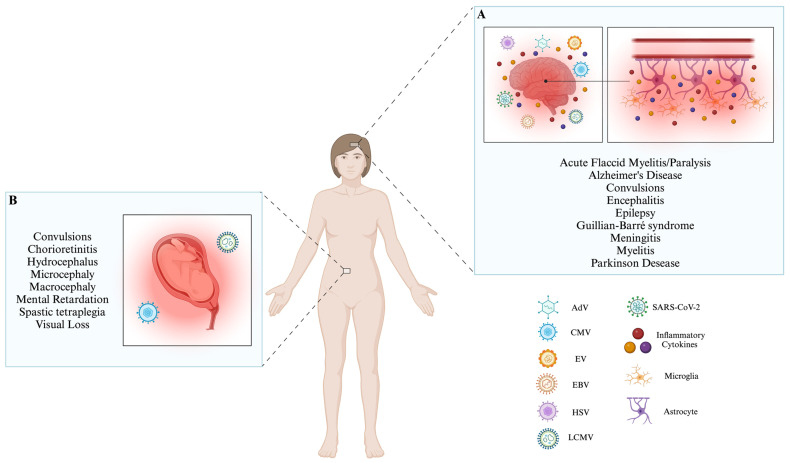
**Infection with neurotrophic viruses leads to several conditions that alter the CNS.** (**A**) The neurotrophic viruses mentioned (AdV, CMV, EV, EBV, HSV, LCMV, and SARS-CoV-2) can enter the CNS via potentially different ways; these include via the nasal epithelium, CSF, via hematogenous spread, through nerves, and/or immune cells. When these viruses interact with the CNS, microglia and astrocytes can secrete inflammatory cytokines, including TNF-α, IL-1β, IL-6, and CCL2. Activating this inflammatory process initiates a range of neurological disorders commonly observed across various types of viruses. (**B**) The presence of LCMV and CMV infections during pregnancy poses significant risks to the proper development of the fetus. Both viruses can cause congenital infections, leading to the improper development of the fetus’s central nervous system. Consequently, the newborn may experience significant long-term effects, such as behavioral or physiological abnormalities (Created by BioRender; License # RU26AQJL7Z).

**Figure 2 brainsci-14-00059-f002:**
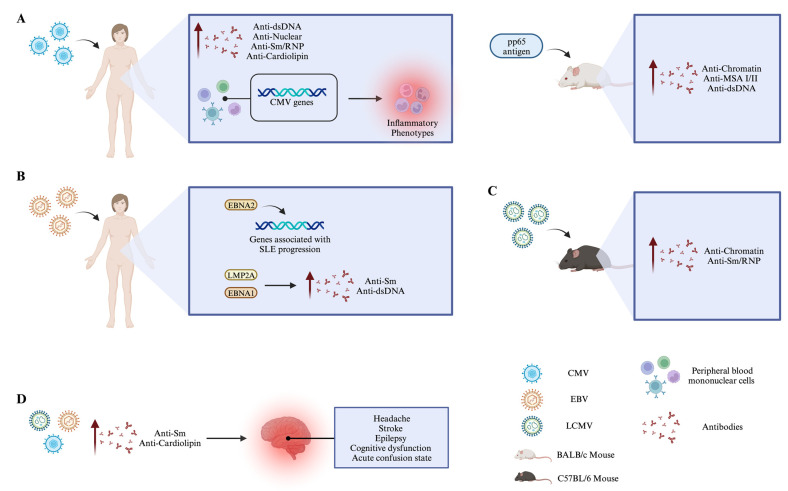
**CMV, EBV, and LMCV are potential effector agents in developing SLE and NPSLE.** (**A**) Infection with CMV in patients who do not have a history of autoimmune diseases can induce the production of anti-nuclear antibodies, anti-dsDNA, anti-Sm/RNP, and anti-cardiolipin antibodies. Furthermore, such infection may decrease complement levels and induce proteinuria. Additionally, 11 CMV genes (UL32, UL36, UL44, UL50, UL56, UL82, UL84, UL95, UL105, UL117, and US31) have been observed in PBMCs from SLE patients, inducing inflammatory phenotypes. Moreover, in BALB/c mice, the administration of the pp65 CMV antigen induces the production of anti-chromatin, anti-centriole, anti-mitotic spindle type I/II (MSA I/II), and anti-dsDNA antibodies. (**B**) The EBV EBNA2 protein can modulate the CD40, LYST, JAZF1, IRF5, BLK, IKZF2, IL12RB2, FAM167A, PTPRC, and SLC15A genes. All of them are associated with the development of SLE. Moreover, LMP2A and EBNA1 proteins induce the production of anti-Sm and anti-dsDNA antibodies. (**C**) LCMV infection in the early stages in the C57BL/6 model produces the generation of autoantibodies, especially anti-chromatin and anti-Sm/RNP. (**D**) Infections triggered by CMV, EBV, and LCMV can potentially induce anti-cardiolipin and anti-Sm antibody production, which generates vascular endothelial cell injury, platelet activation, thrombosis, and neurotoxicity. All of these can trigger NPSLE, presenting clinical symptoms such as headache, stroke (cerebral ischemia and/or intracranial embolism), epilepsy, cognitive dysfunction, and an acute confusion state. Red arrows indicate an increase. (Created by BioRender; License # HK268HVM6O).

**Table 1 brainsci-14-00059-t001:** Immune evasion and CNS clinical manifestations caused by neurotrophic viruses.

Virus	Genome	Invasion to CNS	Immune Response Evasion	Clinical Signs	References
AdV	dsDNA	BBB, and Olfactory nerves	IFN-I pathway modulates the innate and adaptive immune response, and affects antigen presentation processes	Aseptic meningitis convulsions, encephalitis, and acute disseminated encephalomyelitis.	[10,11,39,46,47,48,49,50,51,52]
CMV	dsDNA	BBB	Affects antigen presentation processes module phenotypic and functional changes in macrophages.	Aseptic meningitis, polyradiculopathy, myelitis, ventriculitis, dementia, cranial nerve involvement, microcephaly, periventricular calcifications, convulsions, spastic tetraplegia, and hydrocephalus suppress the development of neural stem cells.	[10,11,53,54,55,56,57,58,59,60,61]
EV	ssRNA^−^	BBB, Myeloid cells infected, motoneurons.	IFN-I pathway regulates the cytosolic RIG-I pathway and NLRP3 inflammasome.	Aseptic meningitis, encephalitis, AFM, cognitive impairments in motor skills, problem-solving abilities, socialization, communication, neuronal development delay, ADHD, and epilepsy.	[10,11,35,62,63,64,65,66,67,68,69,70,71,72,73,74]
EBV	dsDNA	BBB	IFN-I pathway, MAVS, and NLRP3 inflammasome induce apoptosis and inhibit DC development, reprogramming B cells.	Aseptic meningitis, encephalitis, transverse myelitis, cerebellitis, ataxia, Guillian–Barré syndrome, Alzheimer’s Disease, and Parkinson’s disease.	[10,11,75,76,77,78,79,80,81,82,83,84,85,86,87,88]
HSV	dsDNA	BBB, trigeminal or olfactory nerves, latent HSV-1 in situ in the brain, and reactivation of initial infection in the periphery.	IFN-I pathway hinders the complement activation, hinders the ability of IgG antibodies, induces DCs apoptosis, and affects antigen presentation processes.	Aseptic meningitis, acute encephalitis, and cognitive impairment.	[10,11,89,90,91,92,93,94,95,96,97,98,99,100,101,102,103,104]
LCMV	ssRNA^−^	BBB	IFN-I production.	Aseptic meningitis, hydrocephalus, chorioretinitis, macrocephaly, microcephaly, chorioretinopathy, spastic quadriparesis, seizures, visual loss, mental retardation, and neurodegenerative manifestations.	[10,11,105,106,107,108,109,110,111,112,113,114]
SARS-CoV-2	ssRNA^+^	BBB, and cranial nerves.	It affects dsRNA recognition via the RIG-I IFN pathway, induces MAVS autophagy, modules the immune chemotaxis, and affects antigen presentation processes.	Encephalitis, encephalopathy, seizures, epilepsy, and Alzheimer’s Disease.	[115,116,117,118,119,120,121,122,123,124,125,126,127,128,129,130]

## Data Availability

The data are not publicly available due to the fact that all of it was extracted from available and published works which can be openly found in the literature.

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
