# Peer review of "Understanding the Neurotrophic Virus Mechanisms and Their Potential Effect on Systemic Lupus Erythematosus Development"

_brainsci, 2024, doi:10.3390/brainsci14010059_

Round 1
Reviewer 1 Report
Comments and Suggestions for Authors
In the manuscript "Unraveling the Interplay: Viral Influence on Central Nervous System Disorders and Lupus Erythematosus Systemic Development", Felipe R. Uribe and colleagues reviewed the central nervous system infection from several viruses and their association with Lupus Erythematosus Systemic Development.
The bibliography is accurate but unfortunatly nothing is novel in this review. Perhaps the authors can improve the manuscript by using tables and more figures to make it more attrative and easy to read. There is no link between the sections.
Moreover, there are a lot more viruses that can cause CNS infection, including measles virus, henipaviruses, bunyaviruses, mumps, japanese encepalitis virus etc. The authors should at least mention them and explain why they did not include these viruses in the review.
Comments on the Quality of English Language
The English can be improved, especially in the abstract. The main text is clear enough to my mind.
Author Response
Answers to Reviewer 1
- Reviewer #1: In the manuscript "Unraveling the Interplay: Viral Influence on Central Nervous System Disorders and Lupus Erythematosus Systemic Development," Felipe R. Uribe and colleagues reviewed the central nervous system infection from several viruses and their association with Lupus Erythematosus Systemic Development.
Answer: We thank the reviewer for each of his comments because they will help us enhance the quality of our article.
- Reviewer #1: The bibliography is accurate but unfortunatly nothing is novel in this review. Perhaps the authors can improve the manuscript by using tables and more figures to make it more attrative and easy to read. There is no link between the sections.
Answer: As requested by the Reviewer, we have added one table and one more Figure (Pages 16-18, 22; Lines: 642, 766-778).
- Reviewer #1: Moreover, there are a lot more viruses that can cause CNS infection, including measles virus, henipaviruses, bunyaviruses, mumps, japanese encepalitis virus etc. The authors should at least mention them and explain why they did not include these viruses in the review.
Answer: As requested by the Reviewer, we mention other viruses and explain why we did not include these viruses (Page 2; Lines 61-67).
We would like to thank the Reviewers and Editors again for the time and effort invested in handling this manuscript. We hope the current revised version of the document is acceptable for publication in Brain Science.
Reviewer 2 Report
Comments and Suggestions for Authors
The authors include in their review the knowledge about viral infections, CNS pathologies, and the immune response against them.
I think a review like this should have more figures with schemes and explanations.
In addition, the references could be more updated, since the science in this field already has new insights.
Author Response
Answers to Reviewer 2
- Reviewer #2: The authors include in their review the knowledge about viral infections, CNS pathologies, and the immune response against them.
Answer: We thank the reviewer for each of his comments because they will help us enhance the quality of our article.
- Reviewer #2: I think a review like this should have more figures with schemes and explanations.
Answer: As requested by the Reviewer, we have added one table and one more Figure (Pages 16-18, 22; Lines: 642, 766-778).
- Reviewer #2: In addition, the references could be more updated, since the science in this field already has new insights.
Answer: As requested by the Reviewer, we mentioned more updated references (Pages 4-5, 11- 12; Lines: 125, 166, 199, 456, 474, 478, 501).
We would like to thank the Reviewers and Editors again for the time and effort invested in handling this manuscript. We hope the current revised version of the document is acceptable for publication in Brain Science.
Reviewer 3 Report
Comments and Suggestions for Authors
Brief Summary
Uribe et.al. describe a very complex and highly trend topic – the relationship between neurotropic viral infections and human autoimmune diseases such as systemic lupus erythematosus (SLE) and neuropsychiatric lupus (NPSLE) manifestations. Currently, the global incidence of SLE is about 40 per 100,000 and represents a huge challenge to global health. The authors have attempted to summarise the information that links some neurotropic viruses to these pathologies. In particular, they have described in detail the primary immune response to infections with adenoviruses, cytomegalovirus, enteroviruses, Epstein-Barr virus, herpes simplexviruses 1 and 2, lymphocytic choriomeningitis virus (LCMV), and SARS-CoV2. However, the authors' presentation seems a bit rambling and lacking in some important details.
The primary immune response to each of the viruses is described in some detail, which is certainly important and useful information. However, in my opinion, more attention should have been paid to the mechanisms of virus penetration into the CNS, associated with increased permeability of the BBB barriers, as well as the mechanisms of cell destruction in the CNS or virus multiplication in the CNS, due to which the lesion will occur.
General comments
The title of the article does not match the content of the article. In this case we see rather a description of the innate immune response for some viruses associated with neurodegenerative diseases. Such an important viral pathogen as flaviviruses, some of which are also neurotropic, has been omitted. It is not noted which of these viruses penetrate the CNS during acute infection, and for which of these viruses these lesions may be associated with a chronic course of the disease.
Specific comments
Line 64-77. The authors tried to introduce us to the section of the article on neurotropic viruses, but they mentioned only the mechanisms of penetration of RNA viruses into the CNS, although in their work, they also describe DNA viruses. We would like to clarify in this section what mechanisms are characteristic for them perhaps to note common features or differences.
Line 83-87. I would like a more detailed description of the adenovirus virions structure. It is not clear why these proteins were chosen in the paragraph. Their role is not clear (which of them are on the surface and are related to virus attachment and penetration, which are related to nucleocapsid formation and binding to DNA).
Line 92-94. The information presented here is erroneous and taken out of context. This phrase gives the impression that the % of neurological complications in children infected with adenoviruses is very high, although this is not true. The original sentence should have said that they mainly cause febrile respiratory illness and in rare cases can cause CNS disease (~3.3%).
Line 135-149. It should also first be pointed out that CMV infection is more often asymptomatic and only in a small number of cases and most commonly in immunocompromised patients.
In the description of CMV, there is no mention of the relationship of the virus with central nervous system disorders and systemic development of lupus erythematosus, or possible mechanisms of development of these pathologies. I'd like to see some small conclusion to this topic after each of the sections.
Line 181-183. Not all enteroviruses cause CNS pathology. I would like some clarification on this point.
Line 184-185. Could you please specify where these processes take place? After penetration into the CNS or before? I would like more details for clarity.
Line 295-304. Write in what % of cases the HSV infection is associated with neurodegenerative diseases, what factors contribute to it.
In conclusion, we could also add the information that the main way to treat these pathologies is immunosuppressive therapy and look at the relevant studies in this direction, as well as indicate how such therapy may affect the course of the infectious process.
Author Response
Answers to Reviewer 3
- Reviewer #3: Uribe et.al. describe a very complex and highly trend topic – the relationship between neurotropic viral infections and human autoimmune diseases such as systemic lupus erythematosus (SLE) and neuropsychiatric lupus (NPSLE) manifestations. Currently, the global incidence of SLE is about 40 per 100,000 and represents a huge challenge to global health. The authors have attempted to summarise the information that links some neurotropic viruses to these pathologies. In particular, they have described in detail the primary immune response to infections with adenoviruses, cytomegalovirus, enteroviruses, Epstein-Barr virus, herpes simplex viruses 1 and 2, lymphocytic choriomeningitis virus (LCMV), and SARS-CoV2. However, the authors' presentation seems a bit rambling and lacking in some important details.
Answer: We thank the Reviewer for each of his comments because they will help us enhance the quality of our article.
- Reviewer #3: The primary immune response to each of the viruses is described in some detail, which is certainly important and useful information. However, in my opinion, more attention should have been paid to the mechanisms of virus penetration into the CNS, associated with increased permeability of the BBB barriers, as well as the mechanisms of cell destruction in the CNS or virus multiplication in the CNS, due to which the lesion will occur.
Answer: As requested by the Reviewer, we have added information about mechanisms and cells (Pages 3, 5-8, 10, 12-14; Lines: 109-116,177-181, 238-250, 318-326, 391-409, 486-494, 506-508, 566-573).
- Reviewer #3: The title of the article does not match the content of the article. In this case we see rather a description of the innate immune response for some viruses associated with neurodegenerative diseases. Such an important viral pathogen as flaviviruses, some of which are also neurotropic, has been omitted. It is not noted which of these viruses penetrate the CNS during acute infection, and for which of these viruses these lesions may be associated with a chronic course of the disease.
Answer: As requested by the Reviewer, we have changed the title, and we explain why we omitted flaviviruses (Pages 1-2; Lines: 1-4, 61-67).
- Reviewer #3: Line 64-77. The authors tried to introduce us to the section of the article on neurotropic viruses, but they mentioned only the mechanisms of penetration of RNA viruses into the CNS, although in their work, they also describe DNA viruses. We would like to clarify in this section what mechanisms are characteristic for them perhaps to note common features or differences.
Answer: As requested by the Reviewer, we include the required information (Page 2-3; Lines: 81-84).
- Reviewer #3: Line 83-87. I would like a more detailed description of the adenovirus virions structure. It is not clear why these proteins were chosen in the paragraph. Their role is not clear (which of them are on the surface and are related to virus attachment and penetration, which are related to nucleocapsid formation and binding to DNA).
Answer: As requested by the Reviewer, we have provided more detailed information (Page 3; Lines: 90-99).
- Reviewer #3: Line 92-94. The information presented here is erroneous and taken out of context. This phrase gives the impression that the % of neurological complications in children infected with adenoviruses is very high, although this is not true. The original sentence should have said that they mainly cause febrile respiratory illness and in rare cases can cause CNS disease (~3.3%).
Answer: As requested by the Reviewer, we have corrected this sentence (Page 3; Lines: 119-123).
- Reviewer #3: Line 135-149. It should also first be pointed out that CMV infection is more often asymptomatic and only in a small number of cases and most commonly in immunocompromised patients.
Answer: As requested by the Reviewer, we have corrected this point (Pages 5; Lines: 172-174).
- Reviewer #3: In the description of CMV, there is no mention of the relationship of the virus with central nervous system disorders and systemic development of lupus erythematosus, or possible mechanisms of development of these pathologies. I'd like to see some small conclusion to this topic after each of the sections.
Answer: As requested by the Reviewer, we have included small conclusions at the end of each section (Pages 16, 22; Lines: 628-640, 757-764).
- Reviewer #3: Line 181-183. Not all enteroviruses cause CNS pathology. I would like some clarification on this point.
Answer: As requested by the Reviewer, we specify this point (Page 6; Lines: 235-238).
- Reviewer #3: Line 184-185. Could you please specify where these processes take place? After penetration into the CNS or before? I would like more details for clarity.
Answer: As requested by the Reviewer, we have added more detail to specify this point (Page 6; Lines: 238-241).
- Reviewer #3: Line 295-304. Write in what % of cases the HSV infection is associated with neurodegenerative diseases, what factors contribute to it.
Answer: As requested by the Reviewer, we have added this information (Page 10; Lines: 410-415).
- Reviewer #3: In conclusion, we could also add the information that the main way to treat these pathologies is immunosuppressive therapy and look at the relevant studies in this direction, as well as indicate how such therapy may affect the course of the infectious process.
Answer: As requested by the Reviewer, we have included information about potential new treatment (Page 23; Lines: 805-807).
We would like to thank the Reviewers and Editors again for the time and effort invested in handling this manuscript. We hope the current revised version of the document is acceptable for publication in Brain Science.
Reviewer 4 Report
Comments and Suggestions for Authors
The review article titled "Unraveling the Interplay: Viral Influence on Central Nervous System Disorders and Lupus Erythematosus Systemic Development" focuses on the role of neurotropic viruses in central nervous system (CNS) pathologies, which are a significant public health concern. These viruses are characterized by their ability to infiltrate the CNS, interacting with various cell populations and inducing several diseases. The immune response in the CNS against these viruses is primarily driven by microglia, which secrete inflammatory cytokines to combat the infection. The review highlights the most relevant neurotropic viruses, including adenovirus (AdV), cytomegalovirus (CMV), enterovirus (EV), Epstein-Barr Virus (EBV), herpes simplex virus type 1 (HSV-1) and 2 (HSV-2), lymphocytic choriomeningitis virus (LCMV), and the recently discovered SARS-CoV-2. Additionally, the article discusses the association of viral infections with systemic lupus erythematosus (SLE) and neuropsychiatric lupus (NPSLE), reviewing the current understanding of viral infections, CNS pathologies, and the immune response against them. It also highlights the importance of understanding the role of various viral proteins in developing neuronal pathologies, SLE, and NPSLE​
I have some comments/concerns for this article:
Originality and Significance: The topic of the interplay between viral infections and their influence on central nervous system disorders, as well as systemic lupus erythematosus, is highly relevant and original. The comprehensive review of various neurotropic viruses and their potential roles in these conditions is commendable. It is essential to highlight the implications of these findings for medical research and clinical practice.
Literature Review:
1. The authors have done a thorough job of compiling and discussing relevant literature. However, it might be beneficial to include more recent studies or emerging data, especially related to the ongoing COVID-19 pandemic and its neurological impacts, which could be relevant to the article's scope.
2. Introduction line 37-38 is supported by one reference from 2018. Can authors help this information with more recent data?
3. The prevalence of viruses varies in different geographical locations. Does it impact the CNS pathologies across other regions? Authors are advised to add data.
Methodology and Analysis:
The authors have systematically presented the information. Future research directions or potential methodologies for exploring the connections between viral infections and the discussed disorders could be suggested.
A comparative analysis of the different neurotropic viruses, in terms of their mechanisms of CNS infiltration and immune evasion strategies, which can be presented by figure or table, could enhance the reader's understanding. It would be beneficial if these visual aids were utilised effectively to summarise key points and provide a quick reference for complex information.
Clarity and Structure: The article is well-structured, with clear subheadings and a logical flow. However, some sections could benefit from summaries or conclusions to reinforce key points and enhance readability.
Clinical Relevance: The review's connection to clinical applications is evident but could be strengthened. Discussions on potential diagnostic, therapeutic, and preventative strategies in light of the reviewed literature would be valuable. This would make the article more appealing to clinicians and healthcare professionals.
References: The references are comprehensive and well-cited. I want you to know that ensuring all references are up-to-date and include any recent breakthrough studies would enhance the article's relevance and accuracy.
Comments on the Quality of English Language
Minor editing of the English language required
Author Response
Answers to Reviewer 4
- Reviewer #4: The review article titled "Unraveling the Interplay: Viral Influence on Central Nervous System Disorders and Lupus Erythematosus Systemic Development" focuses on the role of neurotropic viruses in central nervous system (CNS) pathologies, which are a significant public health concern. These viruses are characterized by their ability to infiltrate the CNS, interacting with various cell populations and inducing several diseases. The immune response in the CNS against these viruses is primarily driven by microglia, which secrete inflammatory cytokines to combat the infection. The review highlights the most relevant neurotropic viruses, including adenovirus (AdV), cytomegalovirus (CMV), enterovirus (EV), Epstein-Barr Virus (EBV), herpes simplex virus type 1 (HSV-1) and 2 (HSV-2), lymphocytic choriomeningitis virus (LCMV), and the recently discovered SARS-CoV-2. Additionally, the article discusses the association of viral infections with systemic lupus erythematosus (SLE) and neuropsychiatric lupus (NPSLE), reviewing the current understanding of viral infections, CNS pathologies, and the immune response against them. It also highlights the importance of understanding the role of various viral proteins in developing neuronal pathologies, SLE, and NPSLE​
I have some comments/concerns for this article:
Originality and Significance: The topic of the interplay between viral infections and their influence on central nervous system disorders, as well as systemic lupus erythematosus, is highly relevant and original. The comprehensive review of various neurotropic viruses and their potential roles in these conditions is commendable. It is essential to highlight the implications of these findings for medical research and clinical practice.
Answer: We thank the Reviewer for each of his comments because they will help us enhance the quality of our article.
- Reviewer #4: The authors have done a thorough job of compiling and discussing relevant literature. However, it might be beneficial to include more recent studies or emerging data, especially related to the ongoing COVID-19 pandemic and its neurological impacts, which could be relevant to the article's scope.
Answer: As requested by the Reviewer, we have added related data (Page 14; Lines: 560-566).
- Reviewer #4: Introduction line 37-38 is supported by one reference from 2018. Can authors help this information with more recent data?
Answer: As requested by the Reviewer, we have included recent data (Page 2, line 46).
- Reviewer #4: The prevalence of viruses varies in different geographical locations. Does it impact the CNS pathologies across other regions? Authors are advised to add data.
Answer: As requested by the Reviewer, we have added information about the geographical incidences (Pages 3-6, 8, 10, 12, 14; Lines: 102-109, 169-177, 232-234, 314-317, 388-391, 482-486, 560-566).
- Reviewer #4: The authors have systematically presented the information. Future research directions or potential methodologies for exploring the connections between viral infections and the discussed disorders could be suggested.
Answer: As requested by the Reviewer, we have added information about detection methodologies (Page 23; Lines: 795-805).
- Reviewer #4: A comparative analysis of the different neurotropic viruses, in terms of their mechanisms of CNS infiltration and immune evasion strategies, which can be presented by figure or table, could enhance the reader's understanding. It would be beneficial if these visual aids were utilised effectively to summarise key points and provide a quick reference for complex information.
Answer: As requested by the Reviewer, we have added one more table and Figure (Pages 16-18, 22; Lines: 642, 766-778).
- Reviewer #4: The article is well-structured, with clear subheadings and a logical flow. However, some sections could benefit from summaries or conclusions to reinforce key points and enhance readability.
Answer: As requested by the Reviewer, we have added summaries (Pages 4, 6, 8-15; Lines: 152-156, 215-221, 294-300, 372-377, 457-463, 539-545, 622-627).
Reviewer #4: The review's connection to clinical applications is evident but could be strengthened. Discussions on potential diagnostic, therapeutic, and preventative strategies in light of the reviewed literature would be valuable. This would make the article more appealing to clinicians and healthcare professionals.
Answer: As requested by the Reviewer, we add information about these points (Page 23; Lines: 797-810).
- Reviewer #4: The references are comprehensive and well-cited. I want you to know that ensuring all references are up-to-date and include any recent breakthrough studies would enhance the article's relevance and accuracy.
Answer: As requested by the Reviewer, we have included more updated references (Pages 4-5, 11- 12; Lines: 125, 166, 199, 456, 474, 478, 501).
We would like to thank the Reviewers and Editors again for the time and effort invested in handling this manuscript. We hope the current revised version of the document is acceptable for publication in Brain Science.
Round 2
Reviewer 1 Report
Comments and Suggestions for Authors
The authors took into account my comments and the manuscript is ok for publication to my mind now.
Reviewer 3 Report
Comments and Suggestions for Authors
The authors have attempted to combine what is known about the mechanisms of neurotropic viral infection with a focus on the development of autoimmunity in lupus erythematosus. After the corrections made, the article looks much more logical and understandable. The data presented are useful and can be used by other researchers in the field.
A few minor comments:
Line 96: About the µ protein has been said before, I think it's a repeat
Line 97: Does Mu mean µ protein? –clarify please
Line 117-118: Perhaps it would be better to move this sentence to line 102.

Reviewer 4 Report
Comments and Suggestions for Authors
The authors have answered my questions, and the paper has significantly improved.